# An Integrated Clinical-Biological Approach to Identify Interindividual Variability and Atypical Phenotype-Genotype Correlations in Myopathies: Experience on A Cohort of 156 Families

**DOI:** 10.3390/genes12081199

**Published:** 2021-07-31

**Authors:** Raul Juntas Morales, Aurélien Perrin, Guilhem Solé, Delphine Lacourt, Henri Pegeot, Ulrike Walther-Louvier, Pascal Cintas, Claude Cances, Caroline Espil, Corinne Theze, Reda Zenagui, Kevin Yauy, Elodie Cosset, Dimitri Renard, Valerie Rigau, Andre Maues de Paula, Emmanuelle Uro-Coste, Marie-Christine Arne-Bes, Marie-Laure Martin Négrier, Nicolas Leboucq, Blandine Acket, Edoardo Malfatti, Valérie Biancalana, Corinne Metay, Pascale Richard, John Rendu, François Rivier, Michel Koenig, Mireille Cossée

**Affiliations:** 1Explorations Neurologiques et Centre SLA, Centre de Référence des Maladies Neuromusculaires AOC (Atlantique-Occitanie-Caraïbe), Centre Hospitalier Universitaire de Montpellier, 34295 Montpellier, France; rjuntas@vhebron.net; 2Équipe Accueil EA7402, Institut Universitaire de Recherche Clinique (IURC), Université de Montpellier, 34093 Montpellier, France; elodie.cosset@hotmail.fr; 3Laboratoire de Génétique Moléculaire, Centre Hospitalier Universitaire de Montpellier, 34093 Montpellier, France; aurelien.perrin@ext.inserm.fr (A.P.); d-thorel@chu-montpellier.fr (D.L.); pegeot.henri@gmail.com (H.P.); corinne.theze@inserm.fr (C.T.); mr-zenagui@chu-montpellier.fr (R.Z.); k-yauy@chu-montpellier.fr (K.Y.); michel.koenig@inserm.fr (M.K.); 4PhyMedExp, Université de Montpellier, INSERM, CNRS, 34093 Montpellier, France; 5Service de Neurologie, Centre Hospitalier Universitaire de Bordeaux, Centre de référence des Maladies Neuromusculaires AOC (Atlantique-Occitanie-Caraïbe), 33000 Bordeaux, France; guilhem.sole@chu-bordeaux.fr; 6Service de Neuropédiatrie, Centre Hospitalier Universitaire de Montpellier, Centre de référence des Maladies Neuromusculaires AOC (Atlantique-Occitanie-Caraïbe), 34295 Montpellier, France; u-louvier@chu-motpellier.fr (U.W.-L.); f-rivier@chu-montpellier.fr (F.R.); 7Service de Neurologie, Centre Hospitalier Universitaire de Toulouse, Centre de référence des Maladies Neuromusculaires AOC (Atlantique-Occitanie-Caraïbe), 31059 Toulouse, France; cintas.p@chu-toulouse.fr (P.C.); arne-bes.mc@chu-toulouse.fr (M.-C.A.-B.); acket.b@chu-toulouse.fr (B.A.); 8Service de Neuropédiatrie, Centre Hospitalier Universitaire de Toulouse, Centre de référence des Maladies Neuromusculaires AOC (Atlantique-Occitanie-Caraïbe), 31059 Toulouse, France; cances.c@chu-toulouse.fr; 9Service de Neuropédiatrie, Centre Hospitalier de Bordeaux, Centre de référence des Maladies Neuromusculaires AOC (Atlantique-Occitanie-Caraïbe), 33000 Bordeaux, France; caroline.espil@chu-bordeaux.fr; 10Service de Neurologie, Centre Hospitalier Universitaire de Nîmes, Centre de référence des Maladies Neuromusculaires AOC (Atlantique-Occitanie-Caraïbe), 30029 Nîmes, France; dimitri.renard@chu-nimes.fr; 11Service de Pathologie, Centre Hospitalier Universitaire de Montpellier, Centre de Référence des Maladies Neuromusculaires AOC (Atlantique-Occitanie-Caraïbe), 34295 Montpellier, France; v-rigau@chu-montpellier.fr; 12Service de Pathologie, Centre Hospitalier Universitaire de Marseille, Centre de Référence des Maladies Neuromusculaires PACA-Réunion-Rhône Alpes, 13005 Marseille, France; andre.mauesdepaula@ap-hm.fr; 13Service de Pathologie, Centre Hospitalier Universitaire de Toulouse, Centre de Référence des Maladies Neuromusculaires AOC (Atlantique-Occitanie-Caraïbe), 31300 Toulouse, France; uro-coste.e@chu-toulouse.fr; 14CHU de Bordeaux, Institut des Maladies Neurodégénératives, Université de Bordeaux, UMR 5293, 33076 Bordeaux, France; marie-laure.martin-negrier@chu-bordeaux.fr; 15Service de Neuroradiologie, Centre Hospitalier de Montpellier, Centre de Référence des Maladies Neuromusculaires AOC (Atlantique-Occitanie-Caraïbe), 34295 Montpellier, France; n-leboucq@chu-montpellier.fr; 16Service Neurologie Médicale, Centre de Référence Maladies Neuromusculaires Nord-Est-Ile-de-France, CHU Raymond-Poincaré, 92380 Garches, France; edoardo.malfatti@aphp.fr; 17U1179 UVSQ-INSERM Handicap Neuromusculaire: Physiologie, Biothérapie et Pharmacologie Appliquées, UFR des Sciences de la Santé Simone Veil, Université Versailles-Saint-Quentin-en-Yvelines, 78180 Versailles, France; 18Laboratoire de Diagnostic Génétique, Université de Strasbourg, 67084 Strasbourg, France; valerie.biancalana@chru-strasbourg.fr; 19Institut de Génétique et de Biologie Moléculaire et Cellulaire (IGBMC), Inserm U1258, CNRS UMR7104, Université de Strasbourg, 67404 Illkirch, France; 20Unité Fonctionnelle de Cardiogénétique et Myogénétique, Centre de Génétique, Hôpitaux Universitaire Pitié Salpêtrière–Charles Foix, 75651 Paris, France; corinne.metay@aphp.fr (C.M.); pascale.richard@aphp.fr (P.R.); 21CHU Grenoble, Université de Grenoble Alpes, Inserm, U1216, GIN, 38706 Saint-Martin-d’Hères, France; JRendu@chu-grenoble.fr; 22Unité Médicale de Génétique Moléculaire, Centre Hospitalier, Universitaire Grenoble Alpes, 38043 Saint-Martin-d’Hères, France

**Keywords:** myopathies, next generation sequencing, deep phenotyping, inter-individual variability, atypical phenotype-genotype associations

## Abstract

Diagnosis of myopathies is challenged by the high genetic heterogeneity and clinical overlap of the various etiologies. We previously reported a Next-Generation Sequencing strategy to identify genetic etiology in patients with undiagnosed Limb-Girdle Muscular Dystrophies, Congenital Myopathies, Congenital Muscular Dystrophies, Distal Myopathies, Myofibrillar Myopathies, and hyperCKemia or effort intolerance, using a large gene panel including genes classically associated with other entry diagnostic categories. In this study, we report the comprehensive clinical-biological strategy used to interpret NGS data in a cohort of 156 pediatric and adult patients, that included Copy Number Variants search, variants filtering and interpretation according to ACMG guidelines, segregation studies, deep phenotyping of patients and relatives, transcripts and protein studies, and multidisciplinary meetings. Genetic etiology was identified in 74 patients, a diagnostic yield (47.4%) similar to previous studies. We identified 18 patients (10%) with causative variants in different genes (*ACTA1, RYR1, NEB, TTN, TRIP4, CACNA1S, FLNC, TNNT1,* and *PAPBN1*) that resulted in milder and/or atypical phenotypes, with high intrafamilial variability in some cases. Mild phenotypes could mostly be explained by a less deleterious effect of variants on the protein. Detection of inter-individual variability and atypical phenotype-genotype associations is essential for precision medicine, patient care, and to progress in the understanding of the molecular mechanisms of myopathies.

## 1. Introduction

Inherited myopathies are clinically and genetically heterogeneous diseases, with 13 clinical and/or histological entry diagnosis groups reported in the Gene Table of Neuromuscular Disorders, (http://www.musclegenetable.fr, accept on 9 July 2021) [1] More than 200 genes are implicated including the giant and complex titin (*TTN*) and nebulin (*NEB*) genes. The complexity increases with the evolution of knowledge and the identification of important clinical overlaps between each phenotypic entry. The boundaries of the phenotypic entries reported in the classification of myopathies are thus becoming increasingly ambiguous [2].

Next-generation sequencing (NGS) technologies have emerged as a rapid approach to simultaneously analyze multiple genes, including very large genes such as ***TTN*** and ***NEB***. Previous NGS studies in muscle diseases showed a variable rate of genetic diagnosis, ranging from 30 to 50% [3,4,5,6,7,8], depending on the number of genes included in the panel and the myopathy classification criteria. 

We previously reported the implementation of a targeted NGS strategy and a variant prioritization tool, the MoBiDiC prioritization algorithm (MPA) to identify variants in patients with undiagnosed Limb-Girdle Muscular Dystrophies (LGMD), Congenital Myopathies (CM), Congenital Muscular Dystrophies (CMD), distal myopathies (DM), Myofibrillar Myopathies (MFM) and hyperCKemia or effort intolerance [9,10]. Because of the clinical overlap between phenotypic groups, we added additional genes associated with other entry diagnostic categories (mainly Congenital Myasthenic Syndromes (CMS), retractile myopathies, and metabolic myopathies) [9], to search for potential atypical phenotype-genotype associations. We also updated the gene panel to include newly identified genes, leading to a panel of 185 genes (Appendix A). Compared to whole-exome sequencing, gene panel sequencing provides higher read depth and hence better coverage of all exons, which in addition is useful for better detection of CNV (deletions or duplications), as we previously demonstrated for large genes such as nebulin or titin [9]. In this study, we report the comprehensive clinical-biological approach we used to interpret NGS data in a cohort of 156 patients and showed that this strategy is essential to progress in the understanding of the molecular mechanisms of myopathies, by identifying inter-individual variability and atypical phenotype-genotype correlations in myopathies.

## 2. Materials and Methods

### 2.1. Patients

DNA samples were from a cohort of 156 patients (82 men and 74 women) followed at the French South-West Reference Center for Neuromuscular Disorders. This cohort included 88 patients with the pediatric-onset disease (neonatal onset in 36 patients) and 68 adults, all with a genetically unclassified myopathy. Most cases were sporadic. Family history was reported by 22 patients from 20 different families, with a pattern of inheritance that was autosomal dominant (AD) in 12 families, autosomal recessive (AR) in 6, and X-linked in 2 families. In 11 patients, a family history of myopathy was revealed *a posteriori* by relatives’ clinical and molecular evaluations. 

Based on their clinical phenotype, patients had LGMD (*n* = 45 patients), CM (*n* = 76), CMD (*n* = 9), DM (*n* = 13), and hyperCKemia or effort intolerance (*n* = 13). Clinical, biochemical, and, when available, radiological and histopathological data were retrieved for all subjects. Most patients underwent previous phenotype-based genetic tests that excluded variations in genes frequently involved in myopathies.

The informed consent was signed by all adult patients and the parents (or legal representatives) for children before molecular genetic testing. The study was approved on 20 July 2021, by the Institutional Review Board of Montpellier University Hospital, with the IRB project identification code: IRB-MTP_2021_07_202100912.

### 2.2. Molecular Analyses

Targeted NGS analyses were performed on DNA using a customized panel of 135 genes (*n* = 133 patients) secondarily increased to 185 genes (*n* = 23 patients) (Appendix A), the SeqCap EZ Choice library capture kit (Roche-NimbleGen), and paired-end sequencing (2 × 150 bp) on a V3 flow-cell using a MiSeq sequencer (Illumina), as previously reported [9]. Variants identified as possibly pathogenic were confirmed by classical Sanger sequencing. For variants predicted to affect splicing, cDNA analyses of the specific gene were performed using mRNA extracted from muscle biopsies, as previously reported [9].

### 2.3. Bioinformatics

Bioinformatics analyses for single nucleotide variant (SNV) and copy number variant (CNV) detection were performed as previously described [9]. The generated variant call files (vcf) were analyzed using an in-house annotation software that is based on ANNOVAR [11] and that uses the following databases for variant annotation: RefGen, Knowngene, clinVar, Kaviar, ExAC03, GnomAD Exome, snp138, dbsnfp, dbscsnv, and spidex. 

### 2.4. Variant Pathogenicity Evaluation

Variants were expected to be present in less than 1% of general population databases such as the Exome Aggregation Consortium (EXAC) and the Genome Aggregation Database (GnomAD). The pathogenicity of the identified variants was assessed using the MPA software [10] and a set of criteria reported by Zenagui et al. 2018 [9], according to the American College of Medical Genetics and Genomics (ACMG) guidelines [12]. The MPA software provides a prediction score of increasing pathogenicity, where 0 predicts a neutral effect and 10 a pathogenic effect.

All probable disease-causing variants were then discussed in multidisciplinary meetings to evaluate their pathogenicity by comparison with the patient phenotype and the suspected mode of inheritance. Further analyses to establish a diagnosis and define genotype-phenotype correlations included familial segregation studies and detailed phenotype analysis through additional clinical examination, whole-body muscle MRI, and retrieval of data concerning biochemical tests and muscle biopsy findings (histology, immunolabeling, and/or electron microscopy). 

## 3. Results

Pathogenic or likely pathogenic variants were detected in 74 of the 156 patients (47.4%) (Appendix A). No disease-causing variant was identified in 72 patients, and a variant of unknown significance (VUS) was detected in 10 patients. 

Among the 98 pathogenic or likely pathogenic variants, 36 were described in the literature as pathogenic, 62 had never been previously reported (Figure 1A). Most of them were missense variants (59.2%) (Figure 1B). 26 different genes were affected, reflecting the genetic diversity of hereditary myopathies, with *RYR1*, *NEB,* and *TTN* the most frequent implicated genes (Figure 1C).

We could perform variants segregation analyses in 58 families. The variant was *de novo* in 13 sporadic cases, mainly *ACTA1* (*n* = 4), *FLNC* (*n* = 2), and *RYR1* (*n* = 2), representing 17.6% of all sporadic cases with a positive genetic test. 

We performed cDNA studies to analyze the effects on transcripts of *CAPN3*, *TTN,* and *NEB* variants predicted to affect splicing [9]. This procedure was particularly important to characterize pathogenicity in complex genes, such as *TTN* and *NEB*. We identified exon skipping in transcripts carrying the splicing variant, and also activation of exonic or intronic cryptic splice sites, leading to aberrant transcripts (Figure 2, and [13]).

### 3.1. Diagnostic Efficiency in Each Phenotypic Group

We reached a definitive diagnosis for 38 of the 76 patients with CM. We found pathogenic variants in 12 genes (*RYR1*, *NEB*, *ACTA1*, *TTN*, *MTM1*, *COL6A1*, *COL6A3*, *MYH7*, *TNNT1*, *CACNA1S*, *TRIP4*, and *FLNC*). In 32 of these patients, the disease had a neonatal onset. The rate of positive diagnosis was particularly high (82%) in patients younger than one year with severe neonatal hypotonia and respiratory insufficiency. Remarkably, only one-third of muscle biopsies performed in patients with the neonatal-onset disease showed specific histologic abnormalities. 

Among the nine patients with CMD (all sporadic cases), we identified the disease cause in five patients who harbored pathogenic variants of the *LAMA2*, *LMNA,* and *FKRP* genes.

In patients with LGMD (*n* = 45), the genetic diagnosis was possible in 18 patients (40%). We detected pathogenic variants in 14 different genes (*DYSF*, *CAPN3*, *FKRP*, *COL6A2*, *COL6A3*, *RYR1*, *ANO5*, *LAMA2*, *DMD*, *SCGG*, *GMPPB*, *MTM1*, *TTN*, and *PABPN1*). All patients underwent a muscle biopsy showing histological features of myopathy or dystrophy. Standard immunohistochemistry studies were performed in 80% of patients, and western blot (WB) analysis only in 30%. As expected, we detected variants in classical LGMD genes mainly in patients without comprehensive immunolabelling analyses on muscle biopsy. In one patient (I24), we identified two variants in the *GMPPB* gene. WB performed after the identification of this variant showed a glycosylation defect in alpha-dystroglycan, thus confirming the implication of the *GMPPB* variants in the patient’s pathology [14]. 

In 8 of the 13 patients with DM, we found a pathogenic variant in four different genes (*NEB*, *TTN*, *MYOT*, and *MYH7*). Of special interest was the finding of two pathogenic *NEB* variants in three adult patients with isolated distal weakness (see below).

Among the 13 patients with isolated hyperCKemia or effort intolerance, we detected pathogenic variants in five patients. Interestingly, variants were in genes usually involved in CM (*ACTA1*), or LGMD (*ANO5*, *D*MD, and *SCGA*) in patients with a pseudo-metabolic phenotype. One patient harbored a previously reported pathogenic variant in the *STIM1* gene.

In our cohort, many patients had a *TTN* gene variant. We found five patients with two pathogenic nonsense, frameshift, and/or splicing *TTN* variants *in trans* (Appendix A). The clinical picture was heterogeneous, varying from congenital centronuclear myopathy to congenital distal arthrogryposis or limb-girdle muscular weakness with cardiomyopathy.

### 3.2. Atypical Phenotype-Genotype Associations

In 51 patients, the pathogenic variant was in one of the genes associated with the entry-diagnosis phenotype, whereas 18 patients had a new or very unusual phenotype/genotype association (Table 1). In some cases, this contributed to expanding the phenotype associated with a specific gene. In the others, the disease severity or the inheritance mode differed from the classical clinical descriptions.

#### 3.2.1. Expanded Phenotype-Genotype Associations

We identified a pathogenic variant in the *RYR1* gene in two siblings (I152 and his sister) with the typical LGMD phenotype. Yet, *RYR1* is one of the genes most frequently implicated in CM but is usually not associated with muscular dystrophy. In the two siblings who developed proximal lower limb weakness in the third decade of life, creatine kinase (CK) level was very high (x20) and muscle biopsy showed dystrophic features (Figure 3e-h). The homozygous missense variant identified in the *RYR1* gene, c.6617C>T; p. (Thr2206Met) (SIFT: 0.00; PolyPhen2: 0.998; MPA score: 9), was previously described in malignant hyperthermia with AD transmission. This variant has been found, infrequently, also in patients with typical central core disease (AR transmission), either in a compound heterozygous state in association with truncating variants [14], or in a homozygous state [15].

Moreover, we identified a homozygous in-frame deletion of exon 8 and 9 in the TRIP4 gene in an adult patient (I60) with CM from a consanguineous family. TRIP4 encodes one of the subunits of the tetrameric transcriptional coactivator Activating Signal Co-integrator 1 (ASC-1). In 2016, truncating TRIP4 variants (AR transmission) were implicated in severe congenital myopathies [15] and spinal muscular atrophy [16]. Patient I60 had a completely different and milder phenotype with slowly progressive mild proximal and axial weakness since childhood, associated with later-onset dilated cardiomyopathy (Table 1). Western blot analysis of ASC-1 expression in the muscle biopsy showed the absence of the full-length protein [17]. In this patient, we also found a truncating variant in the TTN gene (c.6379_6380del; p.(Tyr2127Leufs*8). This emphasizes the complexity and the challenge of NGS data interpretation. Indeed, heterozygous truncating variants in TTN have been mainly implicated in cardiomyopathy without skeletal muscle involvement. The presence of the same TTN variant in his older asymptomatic sister identified by segregation studies also suggested the absence of pathogenicity of this variant alone, at least in the context of AD transmission. 

Finally, we found a (GCN)11 expansion in the PABPN1 gene in a woman with isolated camptocormia (I14). Unlike the typical presentation of oculopharyngeal muscular dystrophy (OPMD) this patient never complained of ptosis or difficulty swallowing. The muscle biopsy showed features compatible with myofibrillar myopathy and rimmed vacuoles. In a recent series, only 6 of the 354 patients with OPMD had a (GCN)11 allele [18]. All these patients had later disease onset and ptosis could be absent. To our knowledge, this is the first case of isolated camptocormia caused by this expansion.

#### 3.2.2. Mild Phenotypes

We identified two likely pathogenic variants in the *NEB* genes in three adults with pure DM: one patient with sporadic disease (I112) and two siblings (I76, I77) in another family. *NEB* variants are usually associated with childhood-onset CM with axial and proximal weakness. Conversely, these three patients had a late-onset prominent distal weakness, with isolated bilateral foot drop (Figure 3c,d). Muscle biopsy analysis by optical and electron microscopy did not find any rod. Patients I76 and I77 had a frameshift deletion in exon 63 (c.8860delG) and a missense variant *in trans*, c.21928T>C; p.(Ser7310Pro), predicted to alter nebulin conformation because proline is known to disrupt protein folding [19]. Patient I112 was compound heterozygous for the missense variants c.21790G>C; p.(Asp726His) (SIFT: 0.001; PolyPhen: 0.897; MPA score: 6) and c.194C>T; p.(Pro65Leu) (SIFT: 0.00; PolyPhen: 0.32; MPA score: 4). Despite the inconclusiveness about the pathogenicity of the *NEB* variants in this patient, we hypothesized that missense *NEB* variants could be responsible for a milder adult-onset phenotype without rods, as reported by Wallgren-Petterson et al. [20]. These authors described four Finnish families with two different homozygous *NEB* missense variants and mild distal myopathy. Comparison of the variants found in our three patients with some previously reported *NEB* missense variants indicated that they predominantly occur at conserved residues in the *NEB* repeat motif (Dx2(E/D)x4Kx6(S/N)x3YK), and that proline mutants cluster at specific sites of this motif (Figure 4).

Similarly, NGS allowed the identification of a *de novo* variant in *ACTA1* [c.889G>A; p.(Ala297Thr)] in a 20-year-old patient (I192) with asymptomatic CK level elevation and normal neurological examination. This missense *ACTA1* variant is not found in the general population (ExAC, gnomAD) (Appendix A). The p.(Ala297Thr) variant is predicted by the FoldX algorithm to induce a shift of Lysine 326 and Lysine 328, due to the size and polarity of the mutant threonine side chain (Figure 5). The change in protein conformation could have an impact on protein function, as Lysine 328 is predicted to interact with the myosin head Glu354 residue. This shift induced by the variant could thus weaken or modify the interaction with the myosin chain. A variant in Lys328 was also identified as responsible for nemaline myopathy suggesting an important role of this residue for muscle contraction [21,22]. Mild adult-onset myopathy has been related to ACTA1 variants [23,24]; however, this is the first description of a subclinical phenotype characterized by asymptomatic CK elevation. However, we cannot exclude that this might represent an age-dependent penetrance issue.

#### 3.2.3. Recently Identified Phenotype-Genotype Associations

We found *FLNC* variants in two patients with CM and cardiomyopathy (I111 and I298). Both patients had a similar phenotype with congenital hypotonia, scoliosis, mild proximal and axial weakness, and early-onset severe cardiomyopathy (Figure 3A). Patient I111 had an *FLNC* variant in the canonical donor splice site of exon 28 (c.4927+2T>A) that is predicted to affect splicing and is not present in the general population. Patient I298 carried a missense variant in exon 21 [c.3557C>T; p.(Ala1186Val)], previously reported as pathogenic [28]. At the time of the identification of these variants, FLNC variants had been associated with cardiomyopathy, but not with CM. This changed in 2017, when Janin et al described a 10-year-old girl with CM, dysmorphic short neck, cardiomyopathy, and reducing bodies in muscle biopsy due to a *de novo* FLNC variant [p.(Gly1168Asp)] (Neuromuscular disorders, Suppl 2, October 2017). More recently, Kiselev et al., [28] reported four patients with very early onset CM (mean age at onset: 2.2 years) with proximal weakness, arthrogryposis (in three patients), and restricted cardiomyopathy. Three of them had the same *FLNC* variant as one of our patients [p.(Ala1186Val)].

Another example of patients with a recently identified phenotype-genotype association concerned two adult patients (I303 and I164) with mild congenital myopathy due to pathogenic variants in the *CACNA1S* gene. Classically, this gene was involved in dominant hypokalemic periodic paralysis but, at the time of our analyses, Schartner et al., [29] reported a series of eleven patients with congenital myopathy due to *CACNA1S* pathogenic variants with recessive or dominant effects. Our two patients presented phenotypic features similar to those described by Schartner et al., particularly severe neonatal hypotonia, respiratory and feeding difficulties, and progressive improvement with age. We confirmed the AR transmission inpatient I303 who harbored two *in trans* nonsense CACNA1S variants [c.2970G>A; p.(Trp990*) and c.5104C>T; p.(Arg1702*)]. We found the missense variant c.2447T>G; p.(Leu816Arg) inpatient I164. This *CACNA1S* variant, which is absent in the general population, affects a very conserved amino acid residue and is predicted to be pathogenic by SIFT (score: 0.0) and PolyPhen2 (score: 0.99). The MPA score was 10. The variant was transmitted by the father, who was paucisymptomatic. Familial segregation analysis revealed the presence of the variant in his father, initially reported as asymptomatic. Finally, a detailed clinical examination showed mild clinical signs and MRI highlighted fatty replacement in the tongue (Figure 3B2), supporting that the variant was pathogenic, but with variable expressivity.

We identified *TNNT1* variants in two patients (I401 and I172). Patient I401 had a homozygous deletion in the *TNNT1* gene (c.192+244_388-1191del), predicted to lead to the in-frame deletion of exons 8 and 9. Western blot analysis revealed the total absence of the troponin protein, no smaller band corresponding to the truncated protein was visible [30]. She had a severe phenotype (marked neonatal hypotonia, rods in the muscle biopsy) that was reminiscent of the previously reported patients with recessive pathogenic variants in *TNNT1* [30,31]. The second patient (I172) had a milder phenotype with proximal and axial weakness in childhood. Neurological examination showed mild dorsal flexion weakness of the feet, *scapula alata*, and facial paresis. We identified a *de novo* missense dominant variant: c.200A>G; p.(His67Arg) in exon 8 (SIFT: 0.002; PolyPhen: 0.22; MPA score: 7). The high MPA score, the absence of this *de novo* variant in the general population (Exac and gnomAD), and its localization in the domain of interaction with tropomyosin support its pathogenic effect. At the time of analysis, only recessive variants in *TNNT1* had been reported (see Discussion).

## 4. Discussion

Our integrated clinical-biological approach to analyze NGS data in a cohort of 156 patients with genetically unclassified myopathies allowed the identification of the genetic defect in 46.7% of patients, and to reveal interindividual variability and atypical phenotype-genotype associations.

The global diagnosis rate is similar to that of previous studies on myopathies (from 30 to 50% [3,4,5,6,7,8,32]) (Figure 1A), whereas preliminary tests (immunolabelling, sequencing of few genes) had excluded the most common muscle disorders, thus constituting a selection bias. This was obvious for LGMD patients in whom muscle immunohistochemical analyses and sequencing of few genes (mainly *CAPN3*, *ANO5,* and *FHSD*) were usually performed as a first-line, probably explaining the lower rate of positive diagnosis (40%) than the other studies (51.2% in the series by Savarese et al., [6]). In most patients where we found LGMD gene variants, retrospective analysis of the muscle biopsy showed that immunolabelling studies were incompletely or not performed. Interestingly, we detected variants in genes that are not usually reported in LGMD (*LAMA2*, *RYR1,* or *PAPBN2*). Unlike other studies, we did not detect any GAA gene variant because, in France, maltase acid activity is measured systematically in most patients suspected to have a myopathy before genetic testing. On the other hand, the genetic diagnosis rate was particularly high (82%) in patients with severe neonatal congenital myopathy and respiratory insufficiency who showed marked genetic heterogeneity. The clinical phenotype is of limited value for the diagnosis in these patients because most neonates with congenital myopathy share common clinical features of hypotonia, poor feeding, and frequently, respiratory insufficiency. Moreover, muscle biopsies performed during the first months of life were often not informative for a specific subtype of myopathy. Consequently, we propose that targeted NGS should be the first diagnostic test for this group of patients to avoid unnecessary muscle biopsies. Furthermore, the high rate of positive results allows us to offer appropriate genetic counseling to the families.

The comprehensive clinical biological diagnostic methodology used in this work could explain the remarkably high percentage of new or unusual phenotype-genotype associations (more than 10% of cases). Indeed, in addition to in silico predictions and familial segregation studies of variants, the procedure for validating the association between atypical or incomplete phenotypes and genetic variants included also deep phenotyping (i.e., detailed clinical examination, whole-body muscle MRI, and additional techniques for muscular biopsy analysis, such as electron microscopy), multidisciplinary concertation, exhaustive and continuous literature update and if necessary, advice from international laboratories that are experts in specific genes. For some cases, functional studies (transcripts and protein) were needed to confirm these correlations (Figure 6). 

Mild phenotypes could be explained by a less deleterious effect of variants on the protein. For instance, in inpatient I14, with a diagnosis of OPMD, the presence of the (GCN)11 expansion in *PABPN1* (only one repeat above the normal limit of ten) might have caused a milder phenotype compared with other patients, due to a less toxic gain-of-function effect. On the other hand, inpatient I192 with asymptomatic CK elevation and a *de novo ACTA1* missense variant p.(Ala297Thr), three-dimensional studies predicted a change in the conformation of alpha-actin that might affect its function by modifying the interaction with the myosin chain, and not by interfering with actin polymerization as reported for classical phenotypes [33,34].

Our study also highlights the importance of clinical and radiological investigations of the patient’s relatives, in order to assess variable expressivity and incomplete penetrance, frequently seen in dominant diseases, as illustrated for patient I164 and his father. The detailed histological description also is fundamental to confirm the deleterious role of a variant. For instance, in patients I142 and I162, NGS analysis identified the presence of a TTN deletion with the dominant transmission in the family. The presence of histological features that are classically associated with *TTN* variants (nuclear internalization and minicores), although not specific, and the typical fatty replacement in the semitendinosus muscle observed by MRI were supplementary evidence in support of the variant pathogenicity (submitted).

To date, only a limited number of patients with nemaline myopathy caused by *TNNT1* variants have been reported in the literature. Initially, the disease was identified in the Amish population in Pennsylvania (homozygous nonsense founder variant c.538G>T; p.(Glu180*)). Later, other AR cases were described in other populations [31,35,36]. We identified *TNNT1* variants in two patients with CM. One of them (I172) had a heterozygous *de novo* missense variant. Recently, Konersman et al., [37] reported a large family with nemaline myopathy caused by a heterozygous missense variant (c.311A>T); p.(Glu104Val) in exon 9 of *TNNT1* that segregates as an AD variant. This clinical description brought some insights into the pathogenic role of the heterozygous variant found in our patient. Moreover, our variant is localized in the same domain of interaction between *TNNT1* and tropomyosin, with a possible effect on the protein affinity. 

Reporting new patients with recently described phenotype-genotype associations in *TNNT1*, *CACNA1S,* and *FLNC* genes is important to better define the clinical features and inheritance pattern of these disorders. Moreover, the availability of more information has highlighted the wide overlap among the clinical phenotypes of different muscular disorders. The NGS technology should significantly contribute to redefining MD classification through extensive gene analysis and detailed phenotype-genotype correlations. In patients without a genetic diagnosis despite targeted NGS analysis, whole exome or whole genome sequencing, possibly complemented by RNAseq studies, will allow identifying the causative variant either in genes already associated with myopathies (e.g., intronic variants) or in genes that have not been implicated in such disorders. A better knowledge of the genetic bases of muscular disorders will contribute to improving our knowledge about the gene and protein functions in normal muscle and pathological conditions. 

## 5. Conclusions

We showed that this approach is essential to progress in the understanding of the molecular mechanisms of myopathies, by identifying inter-individual variability and atypical phenotype-genotype correlations in myopathies. Moreover, our results illustrate that NGS analyses, clinical and radiological examination of relatives are essential to identify pauci-symptomatic relatives and then improve diagnosis and appropriate care in families. 

## Figures and Tables

**Figure 1 genes-12-01199-f001:**
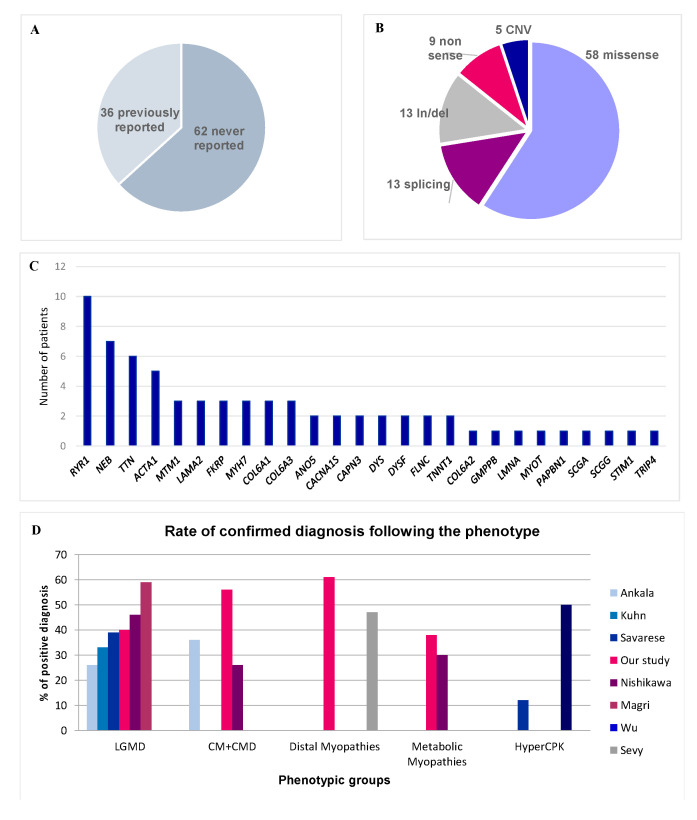
Diagnostic rate and molecular results. (**A**) Reported in literature versus not reported variants. (**B**) Variant type. (**C**) Frequency of pathogenic variants by gene. (**D**) Rate of confirmed diagnosis according to each phenotype in our study in comparison with scientific literature.

**Figure 2 genes-12-01199-f002:**
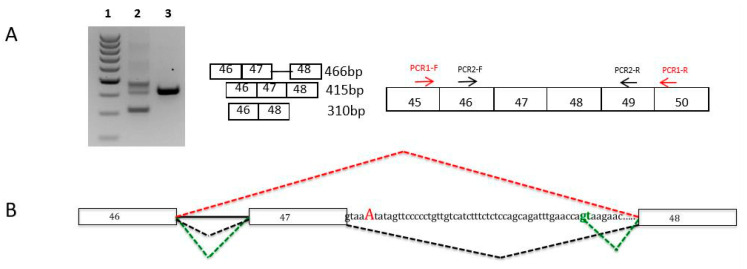
Nebulin muscle transcripts analyses for the c.6075+5G>A variant in intron 47. (**A**) Muscle nebulin transcripts were analyzed by RT-PCR and sequencing from muscle biopsies of the patient harboring the c.6075+5G>A heterozygous variant and from a healthy control. Agarose gel electrophoresis shows the RT-PCR products of nebulin exons 46 to 48 transcripts from the muscle of the patient (Lane 2) and the control (Lane 3). Lane 1 represents the molecular weight marker (100–1000 kb ladder). These analyses showed the presence of three populations of transcripts: a normal product of 415 bases, a smaller transcript (310 bases) corresponding to exon 47 skipping, and a larger fragment (466 bases). This transcript corresponds to the retention of 51 nucleotides in intron 47 due to the inactivation of the WT splice site and the activation of a cryptic donor splice site in intron 47 (green gt). (**B**) Diagram of the splicing pattern in muscle patient. *NEB* exons are shown as boxes. The splicing pattern of the WT RNA is schematically shown in black color and other mutated transcripts in red (exon 47 skipping) and green (aberrant exon 47 skipping).

**Figure 3 genes-12-01199-f003:**
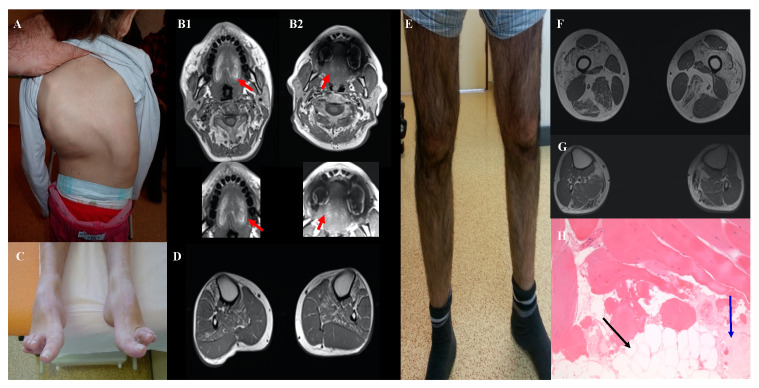
Clinical features of patients with new genotype/phenotype associations: Severe scoliosis in patient I111 aged 5 years with FLNC variant (**A**). Tongue fatty replacement inpatient I164 (**B1**) and his father (**B2**) with CACNA1S variant. Hanging big toe and bilateral tibialis anterior and soleus fatty replacement inpatient I76 with two NEB variants (**C, D**). Patient I152 with LGMD phenotype and homozygous RYR1 variant with severe quadriceps atrophy (**E**), fatty replacement of quadriceps, adductors, semitendinosus, and gastrocnemius muscles in muscular weighted T1 MRI (**F**) and dystrophic pattern in muscular biopsy (**G, H**). The black arrow shows fatty replacement and the blue arrow, fibrosis.

**Figure 4 genes-12-01199-f004:**
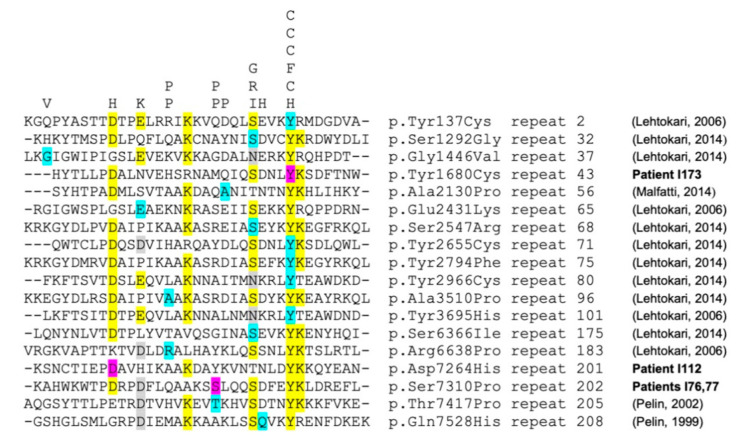
Alignment of *NEB* repeat domains The figure represents the alignment of selected NEB repeats, with conserved residues highlighted in yellow (and/or gray). Residues are indicated by the single-letter amino acid code. “Mutated” residues identified inpatient I112 (p.(Asp7264His)=D7264H, in repeat 201), patients I76,77 (p.(Ser7310Pro)=S7310P, in repeat 202) and patient I173 (p.(Tyr1680Cys)=Y1680C, in repeat 43) are highlighted in purple. “Mutated” residues from previously reported NEB patients with missense variants are highlighted in cyan/blue (references on the right). Corresponding “mutant” residues (single-letter amino acid code) are indicated at the top of the figure. The figure shows that almost all pathogenic missense variants cluster either at the conserved residues of the repeat motif (DX2(E/D)X4KX6(S/N)X3YK) or at 3 specific non-conserved positions for Proline “mutants” (P, at the top of the figure, and which are notorious fold disruptive mutants).

**Figure 5 genes-12-01199-f005:**
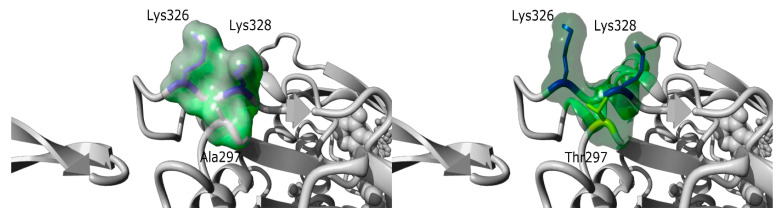
Structure prediction of the p.(Ala297Thr) variant. Based on the 6BNO PDB structures [25] the 297 Alanine residue was mutated in Threonine with the fold X algorithm [26] on YASARA software [27]. The p.(Ala297Thr) variant is predicted by fold X to induce a shift of Lysine 326 and Lysine 328, due to the size and polarity of the mutant threonine side chain. Lysine residues 326 and 328 are shown in dark blue, while Ala297 (left panel) was shown in grey and Threonine 297 (right panel) in yellow. The molecular surface is colored in a green cloud.

**Figure 6 genes-12-01199-f006:**
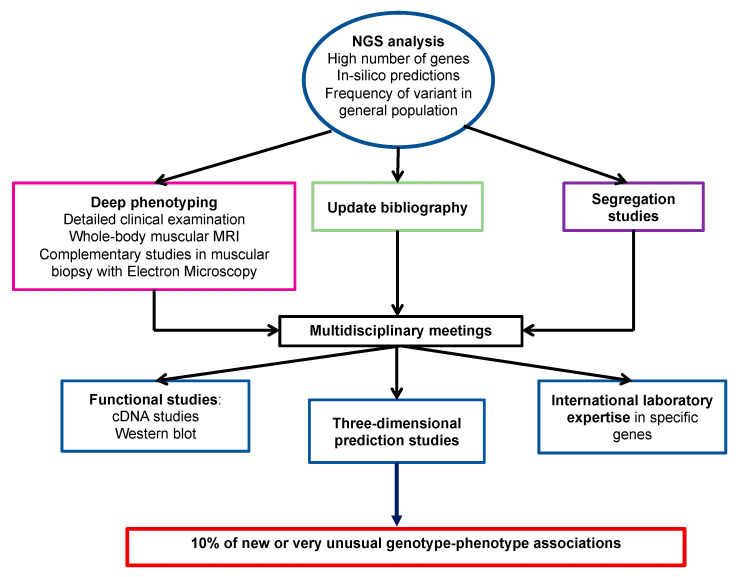
Proposed approach to identify new genotype/phenotype associations.

**Table 1 genes-12-01199-t001:** Clinical features, muscle MRI and histopathological investigations in patients with new or very unusual phenotype/genotype associations.

Patient Sex, Age/ Age at Onset	Clinical Features	CK	Muscle Biopsy	Muscle MRI	Family History	Genetic Testing
**I298** **M, 38y/ neonatal**	Neonatal hypotonia Scoliosis Progressive myopathy with contractures Early-onset cardiomyopathy Restrictive respiratory syndrome	N	Type I fiber predominance. Cytoplasmic bodies	ND	Sporadic	c.3557C>T; p.(Ala1186Val) *FLNC* NM_001458.4 Missense *de novo* variant Absent in Exac and gnomAD
**I111** **F, 4y/ 1y**	Pierre Robin sequence (micrognathia, glossoptosis, and cleft palate), scoliosis Feeding difficulties Severe restrictive cardiomyopathy Mild proximal and axial weakness	N	Type I fiber atrophy	Normal	Sporadic	c.4927+2T>A *FLNC* *de novo* splicing variant HSF, MaxEnt: -100% Absent in Exac and gnomAD
**I192** **M, 20y/ 20y**	Asymptomatic CK elevation Strictly normal neurological examination	x20	Some necrotic fibers Absence of rods	Normal	Sporadic	c.889G>A; p.(Ala297Thr) *ACTA1* NM_001100.3 *de novo* missense variant. Absent in Exac and gnomAD
**I14** **F, 75y/ 60y**	Slowly progressive axial weakness with camptocormia Mild lower limb proximal weakness No ptosis, no dysphagia	N	Abnormal myofibrillar network Rimmed and not rimmed vacuoles	Paravertebral VL, AB, AM, SM, GM, SO	Sporadic	c.30_31insGCA; p.(Ala11dup) *PAPBN1* NM_004643.3
**I303** **F, 30y/ neonatal**	Neonatal hypotonia with feeding difficulties and respiratory insufficiency Progressive improvement Last follow-up visit: mild axial and proximal weakness Gowers + Distal hyperlaxity Worsening with fever	N	Non-specific myopathic pattern with type 1 fiber predominance and mild myofibrillar disorganization	Upper and lower limbs atrophy. No fat tissue replacement	AR (deceased affected brother)	c.2970G>A; p.(Trp990*) *CACNA1S* c.5104C>T; p.(Arg1702*) *CACNA1S* NM_000069.2
**I164** **M, 21y/ neonatal**	Neonatal hypotonia, feeding difficulties, respiratory insufficiency Last follow-up visit: tongue deviation, mild bilateral facial paresis, mild bilateral scapula alata, proximal lower limb weakness, distal hyperlaxity	N	Non-specific myopathic pattern with type 1 fiber predominance	Tongue GMax	Pauci symptomatic father carries the variant	c.2447T>G; p.(Leu816Arg) *CACNA1S* Missense variant predicted to be deleterious Absent in Exac and gnomAD
**I142** **F, 40y/ 35y**	Bilateral calf and left tibialis anterior atrophy	N	Excessive internal nuclei. Minicores with NADH technique	Bilateral GM and left TA	AD	del <11-18> exons *TTN* NM_001267550.1
**I26** **M, 14y/ 3y**	Very mild proximal weakness Mild facial paresis Moderate restrictive respiratory syndrome	N	Excessive internal nuclei resembling centronuclear myopathy. Minicores revealed by NADH staining	Normal	Adopted child	c.51437-4_51444del *TTN* c.26503A>T; p.(Lys8835*) *TTN*
**I47** **F, 44y/ 4y**	Proximal and axial weakness Wheelchair at the age of 17 Last follow-up visit: severe proximal and axial weakness, restrictive respiratory syndrome	N	Excessive internal nuclei resembling centronuclear myopathy. Minicores revealed by NADH staining	Bilateral SO and left TA	Sporadic	c.65575+2T>G *TTN* c.68029A>G; p.(Thr22677Ala) *TTN* Missense variant predicted to affect splicing
**I74, I73** **F, 4y/ 3y**	Mild proximal and axial weakness Facial paresis	x2	Excessive internal nuclei Minicores in NADH staining. Abnormal western blot results: absence of the C-terminal part of TTN (anti-TTN M10-1 antibody); absence of calpain	N.D	Affected twin sister	c.106531G>C; p.(Ala35511Pro) *TTN* Missense variant predicted to affect splicing c.105036C>A; p.(Tyr35012*) *TTN*
**I76, I77** **M, 54y** **F, 50y/ 35y**	Isolated tibialis anterior weakness Recently the brother developed finger extensor and neck flexor weakness	N	Pathological fiber size variation. Excessive internal nuclei Absence of rods	Bilateral TA and GM in both patients	AR	c.8860delG p.(Ala2954Profs*8) *NEB* NM_001271208.1 c.21928T>C; p.(Ser7310Pro) *NEB*
**I112** **M, 60y/ 57y**	Isolated tibialis anterior weakness	N	Pathological fiber size variation. Absence of rods	Bilateral TA and left SO	Sporadic	c.21790G>C p.(Arg7264His) *NEB* c.194C>T p.(Pro65Leu) *NEB*
**I60** **M, 61y/ 20y**	Axial and proximal weakness Retrognathia, pectus excavatum Mild rigid spine Dilated cardiomyopathy	N	Excessive internal nuclei Minicore-like lesions, rods	Bilateral SM, BF, SO	Sporadic (Consang)	del <8-9> homozygous exons *TRIP4* NM_016213.4 In-Frame deletion of exons 8 and 9
**I172** **F, 11y/ neonatal**	Neonatal hypotonia Proximal and axial weakness in childhood. Gowers + Mild tibialis anterior weakness Scapula alata Facial paresis	N	Excessive internal nuclei	Bilateral GMax, AM, SM, BF, TA, SO, P	Sporadic	c.200A>G; p.(His67Arg) *TNNT1* NM_003283.5 *de novo* missense variant Absent in Exac and gnomAD
**I152** **M, 35y/ 20y**	Lower limb-girdle muscular dystrophy	x20	Non-specific dystrophic pattern	Bilateral VL, AM, AB, SM, ST, GM	AR Affected sister. Consang	c.6617C>T; p.(Thr2206Met) Homozygous *RYR1* NM_000540.2 Missense variant

AD: Autosomal Dominant; AR: Autosomal recessive; N: Normal. VL: Vastus lateralis; VM: Vastus medialis; AM: Adductor magnus; AB: Adductor brevis; ST: Semitendinosus; SM: Semimembranosus; BF: Rectus femoris; GMax: Gluteus maximus; GM: Gastrocnemius medialis; TA: Tibialis anterior; P: Peroneus; SO: Soleus; Consang: consanguinity. ND: Not done.

## Data Availability

The data presented in this study are available in the manuscript or in the Appendix A, or can be obtained from the authors upon written request to the corresponding author.

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
