# Peer review of "An Integrated Clinical-Biological Approach to Identify Interindividual Variability and Atypical Phenotype-Genotype Correlations in Myopathies: Experience on A Cohort of 156 Families"

_genes, 2021, doi:10.3390/genes12081199_

Round 1
Reviewer 1 Report
The authors give a thorough report of a multicentric NGS study of patients suffering from myopathies. The main goal is the identification of atypical phenotype-genotype correlations.
It is a pity that the authors changed their customized panel from 135 genes to 185 genes during the analysis process. May be the newest panel could be now applied for the unsolved patients? In table 1 of the supplement a full table of all 185 genes should be shown, may be the newly added genes in a different colour. Please explain why you are still using a customized gene panel and not WES with a selection of genes electronically.
At least in table 2 of the supplement the international classification (ACMG) of variants - class 1 to class 5 - should be applied although the authors have their own in house classification.
Literature seems to me selected and not at the newest status. Literature 9 and 11 is doubled.
Why are the authors using the NMD gene tables for neuromuscular diseases from 2015 and not from this year 2021?
Gene names should be italic. Their is no gene name DYS, it is DMD.
Author Response
The authors give a thorough report of a multicentric NGS study of patients suffering from myopathies. The main goal is the identification of atypical phenotype-genotype correlations.
It is a pity that the authors changed their customized panel from 135 genes to 185 genes during the analysis process. May be the newest panel could be now applied for the unsolved patients? In table 1 of the supplement a full table of all 185 genes should be shown, may be the newly added genes in a different colour. Please explain why you are still using a customized gene panel and not WES with a selection of genes electronically.
We established a panel of neuromuscular genes when we set up NGS in the hospital department. Compared to WES, gene panel sequencing provides higher read depth and hence better coverage of all exons, which in addition is useful for better detection of CNV (deletions or duplications), as we previously demonstrated for large genes such as nebulin or titin (Zenagui et al., 2018 J Mol Diagn.). In addition, in our hospital setting, we only had Illumina MiSeq sequencers, which is not convenient for exome analysis. Within the frame of the French human genome project, unsolved patients will be sequenced by WGS at a later stage; sequencing of the initial patients with the newest panel is therefore not useful. At the initiation of the study, only 135 genes were described as being involved in neuromuscular diseases of genetic origin; the customized gene panel has evolved to 185 genes in accordance with the recommendations (Krahn M. et al. EurJHumGenet 2019). This is now clarified in the manuscript; we have colored in red in Supplementary Table 1 the genes that were added during the update of the NGS gene panel.
At least in table 2 of the supplement the international classification (ACMG) of variants - class 1 to class 5 - should be applied although the authors have their own in house classification.
We have added in Supplementary Table 2, three columns with the CADD score, Splice AI and ACMG classification scores.
Literature seems to me selected and not at the newest status. Literature 9 and 11 is doubled.
We have removed p8 the duplicated paragraph from p8.
Why are the authors using the NMD gene tables for neuromuscular diseases from 2015 and not from this year 2021?
We are sorry for this citation error. We used the 2018 NMD gene tables for neuromuscular diseases. This is now corrected. We did not use the 2021 gene table because the end of this study and of inclusions is end of 2019.
Gene names should be italic. Their is no gene name DYS, it is DMD.
The modifications have been made.
Reviewer 2 Report
This is a nicely written article describing the experiences of a clinical center/laboratory in performing next generation sequencing on patients with myopathies. There are some editing issues that need to be address (see below).
I have a few major points:
- First, it seems the main difference between this paper and their prior work is the addition of a few more genes to their panel. This, in and of itself, is not very novel or interesting. Rather, I find the discussion of their combined approach (deeper phenotyping of family members, segregation studies, molecular modeling, splicing assays) and the atypical/unusual genotype-phenotype correlations the main selling point of this manuscript.
- A major question I would like to see the author’s address is, why perform a panel rather exome sequencing (ES)? With as much as costs have dropped for ES, it is increasingly hard to understand panel-based approaches. For example, panels do not allow novel disease gene identification or identification of pathogenic variants in non-myopathy genes contributing to a blended phenotype. I wonder how many of these “atypical” genotype-phenotype associations are spurious, red herring distracting from the real diagnosis.
- I would like to see CADD and/or REVEL scores for all variants, splicing values for proposed but not tested splice variants (e.g. SpliceAI, MMSplice, etc), and gnomAD frequency. It is hard to interpret variants without this information.
- I would be careful with terminology. This is a major problem in the field. I do not think many of these variants can be regarded as “pathogenic/P” or “likely pathogenic/LP” by ACMG criteria. I might say “candidate variants” instead. You could highlight the variants that are ACMG P/LP in the paper.
- You mention the father of I164 is “paucisymptomatic” in the Results section, which is very vague. You explain what you mean in the Discussion. I would suggest moving the father’s phenotypic/genotypic data to the Results and consider incorporating images from his MRI of the tongue into a figure. Reduced penetrance and variable expressivity are very important points that are underappreciated in the field.
- I found Figure 4 very hard to understand. For example, what do the letters above the alignment mean? Better explanation is needed.
- The ACTA1 variant is very interesting, but it cannot be regarded as definitely pathogenic based on this level of evidence. It is possible this might represent an age-dependent penetrance issue. I would mention this in the paper and again be careful with your language.
- Was there testing for genes linked to Pierre Robin sequence in patient I111? I think this is necessary before attributing it the myopathy gene.
- I appreciate that the gene transcripts are included in Supplemental Table 2. I would consider adding them to the main paper – many readers and even geneticists do not realize how important this is for variant interpretation.
Minor comments:
- Line 79 – Alphadystroglycanopathies should either be hyphenated or two words
- Consider just saying “13 clinical and/or histological entry diagnosis groups” in the first sentence and omit the entities. The sentence is quite long and difficult to read.
- Line 86 - Consider citing a paper describing the blurring of phenotypic boundaries
- All gene names and Latin terms should be italicized (de novo, a posteriori, in trans, etc)
- There is duplication of the first paragraph in the “Expanded phenotype-genotype associations”
- Please indicate when variants are homozygous (missing for the TRIP4 in-frame deletion)
- I would add a citation for the statement that “proline is known to disrupt protein folding” (Line 294)
Author Response
This is a nicely written article describing the experiences of a clinical center/laboratory in performing next generation sequencing on patients with myopathies. There are some editing issues that need to be address (see below).
I have a few major points:
- First, it seems the main difference between this paper and their prior work is the addition of a few more genes to their panel. This, in and of itself, is not very novel or interesting. Rather, I find the discussion of their combined approach (deeper phenotyping of family members, segregation studies, molecular modeling, splicing assays) and the atypical/unusual genotype-phenotype correlations the main selling point of this manuscript.
- A major question I would like to see the author’s address is, why perform a panel rather exome sequencing (ES)? With as much as costs have dropped for ES, it is increasingly hard to understand panel-based approaches. For example, panels do not allow novel disease gene identification or identification of pathogenic variants in non-myopathy genes contributing to a blended phenotype. I wonder how many of these “atypical” genotype-phenotype associations are spurious, red herring distracting from the real diagnosis.
We established a panel of neuromuscular genes when we set up NGS in the hospital department. Compared to WES, gene panel sequencing provides higher read depth and hence better coverage of all exons, which in addition is useful for better detection of CNV (deletions or duplications), as we previously demonstrated for large genes such as nebulin or titin (Zenagui et al., 2018 J Mol Diagn.). In addition, in our hospital setting, we only had Illumina MiSeq sequencers, which is not convenient for exome analysis. Within the frame of the French human genome project, unsolved and atypical patients will be sequenced by WGS at a later stage.
- I would like to see CADD and/or REVEL scores for all variants, splicing values for proposed but not tested splice variants (e.g. SpliceAI, MMSplice, etc), and gnomAD frequency. It is hard to interpret variants without this information.
I would be careful with terminology. This is a major problem in the field. I do not think many of these variants can be regarded as “pathogenic/P” or “likely pathogenic/LP” by ACMG criteria. I might say “candidate variants” instead. You could highlight the variants that are ACMG P/LP in the paper.
We have added in Supplementary Table 2, three columns with the CADD, Splice AI and ACMG classification scores. We agree that according to the ACMG criteria, some variants cannot be classified as "likely pathogenic", but as we pointed out, "deep phenotyping" allowed us to strengthen the pathogenicity of the variant(s) in question. For these reasons, we changed the title of the supplemental table 2 as : Detailed list of pathogenic, likely pathogenic and possibly pathogenic variants.
- You mention the father of I164 is “paucisymptomatic” in the Results section, which is very vague. You explain what you mean in the Discussion. I would suggest moving the father’s phenotypic/genotypic data to the Results and consider incorporating images from his MRI of the tongue into a figure. Reduced penetrance and variable expressivity are very important points that are underappreciated in the field.
We thank the reviewer for this comment. Indeed the description of the father of patient I164 is more suited in the results part, so we moved this text from the discussion part p13 to the result part p11 : “Familial segregation analysis revealed the presence of the variant in his father, initially reported as asymptomatic. Finally, a detailed clinical examination showed mild clinical signs and MRI highlighted fatty replacement in the tongue (Figure 3B2), supporting that the variant was pathogenic, but with variable expressivity.”
p8, we also added in figure 3 the MRI of the father's tongue of patient I164.
We also modified the corresponding sentence in the discussion part, p13:
“Our study also highlights the importance of clinical and radiological investigations of the patient’s relatives, in order to assess variable expressivity and incomplete penetrance, frequently seen in dominant diseases, as illustrated for patient I164 and his father.”
I found Figure 4 very hard to understand. For example, what do the letters above the alignment mean? Better explanation is needed.
Legend of figure 4 was entirely rewritten, as follows:
The figure represents alignment of selected NEB repeats, with conserved residues highlighted in yellow (and/or gray). Residues are indicated by the single-letter amino acid code. “Mutated” residues identified in patient I112 (p.(Asp7264His) =D7264H, in repeat 201), patients I76,77 (p.(Ser7310Pro)=S7310P, in repeat 202) and patient I173 (p.(Tyr1680Cys)=Y1680C, in repeat 43) are highlighted in purple. “Mutated” residues from previously reported NEB patients with missense variants are highlighted in cyan/blue (references on the right). Corresponding “mutant” residues (single-letter amino acid code) are indicated at the top of the figure.
The figure shows that almost all pathogenic missense variants cluster either at the conserved residues of the repeat motif (DX2(E/D)X4KX6(S/N)X3YK) or at 3 specific non-conserved positions for Proline “mutants” (P, at the top of the figure, and which are notorious fold disruptive mutants).
- The ACTA1 variant is very interesting, but it cannot be regarded as definitely pathogenic based on this level of evidence. It is possible this might represent an age-dependent penetrance issue. I would mention this in the paper and again be careful with your language.
According to the reviewer recommendation, we added in p11 this following sentence: “However, we cannot exclude that this might represent an age-dependent penetrance issue.”
- Was there testing for genes linked to Pierre Robin sequence in patient I111? I think this is necessary before attributing it the myopathy gene.
Unfortunately, we have not been able to do any specific test for Pierre Robin syndrome yet. However, we have found similar phenotypes (retrognathia and cleft palate) in 3 others patients with FLNC variants who were previously reported in the literature. (Brodehl et al., 2016 PMID: 26666891) and (Reinstein et al., 2016 PMID: 27601210).
- I appreciate that the gene transcripts are included in Supplemental Table 2. I would consider adding them to the main paper – many readers and even geneticists do not realize how important this is for variant interpretation.
We added the NM gene transcripts in table 1.
Minor comments:
- Line 79 – Alphadystroglycanopathies should either be hyphenated or two words
“Alpha-dystroglycanopathies” has been hyphenated has requested.
- Consider just saying “13 clinical and/or histological entry diagnosis groups” in the first sentence and omit the entities. The sentence is quite long and difficult to read.
The sentence has been modified as requested.
We modified this sentence p1 as follow to introduce abbreviation of phenotypes removed in the first intial sentence: “ We previously reported the implementation of a targeted NGS strategy and a variant prioritization tool, the MoBiDiC prioritization algorithm (MPA) to identify variants in patients with undiagnosed Limb Girdle Muscular Dystrophies (LGMD), Congenital Myopathies (CM), Congenital Muscular Dystrophies (CMD), distal myopathies (DM), Myofibrillar Myopathies (MFM) and hyperCKemia or effort intolerance [9,10].”
- Line 86 - Consider citing a paper describing the blurring of phenotypic boundaries
We added the following reference: “Krahn, M.; Biancalana, V.; Cerino, M.; Perrin, A.; Michel-Calemard, L.; Nectoux, J.; Leturcq, F.; Bouchet-Séraphin, C.; Acquaviva-Bourdain, C.; Campana-Salort, E.; et al. A National French consensus on gene lists for the diagnosis of myopathies using next-generation sequencing. Eur. J. Hum. Genet. 2019, 27, 349–352, doi:10.1038/s41431-018-0305-1.”
- All gene names and Latin terms should be italicized (de novo, a posteriori, in trans, etc)
We thank reviewer 2 for this comment, it was a mistake that occurred during the editorial page layout from the word template. We modified all Latin terms and gene nomenclature in italics.
- There is duplication of the first paragraph in the “Expanded phenotype-genotype associations”
Again, sorry for this mistake linked to page set up.
- Please indicate when variants are homozygous (missing for the TRIP4 in-frame deletion)
We modified this information in table 1.
- I would add a citation for the statement that “proline is known to disrupt protein folding” (Line 294)
We added the following reference in the text p10 “Morgan, A.A.; Rubenstein, E. Proline: The Distribution, Frequency, Positioning, and Common Functional Roles of Proline and Polyproline Sequences in the Human Proteome. PLoS One 2013, 8, e53785, doi:10.1371/journal.pone.0053785.”
Round 2
Reviewer 2 Report
I appreciate the author's responses to my questions and concerns. I feel the responses were adequate.